# Development of a mechatronic weft selector to enhance patterning capacity in Rapier looms

**Aiead Ibne Fahim**[1,2], **Md. Mohaddesh Hosen**[2]*, **Md. Abdullah Al. Mamun**[2,3]*,
**Shah Alimuzzaman Belal**[2]

**1** Department of Textile Engineering, BGMEA University of Fashion & Technology, Dhaka, Bangladesh,
**2** Department of Fabric Engineering, Bangladesh University of Textiles, Dhaka, Bangladesh,
**3** Department of Materials, University of Manchester, Manchester, United Kingdom

☯ These authors contributed equally to this work.
* mohaddesh@fe.butex.edu.bd; almamun@fe.butex.edu.bd

journal.pone.0338603

PORTUGAL

**Peer Review History:** PLOS recognizes the
benefits of transparency in the peer review
process; therefore, we enable the publication
of all of the content of peer review and
author responses alongside final, published
articles. The editorial history of this article is
available here: https://doi.org/10.1371/journal.
pone.0338603

## Abstract

Rapier weaving machines are widely used in the weaving industry; however, the
design of the current weft selectors limits their capability to produce fabrics incorpo-
rating a large variety of weft yarns. Nevertheless, increasing the number of wefts with
the existing selectors can impose challenges for the rapier in the efficient gripping
of the weft. This study presents the design and development of a novel mechatronic
weft selection system aimed at significantly enhancing the patterning capabilities of
rapier looms. The proposed system features a circular arrangement of up to 20 weft
yarn feeders and a single programmable selector module capable of handling the
weft yarns. The selector integrates stepper motors, servo motors, solenoid valves,
and an Arduino Mega-based control unit to execute user-defined weft patterns with
high precision. The device was successfully operated at 9 picks per minute (PPM)
across various yarn types, including polyester, viscose, and elastomeric yarns. A total
of 11 different yarns were tested, each subjected to 5 trials using 20 feeders, result-
ing in the insertion of 1100 wefts. The system achieved a 100% selection and inser-
tion success rate after the trials. The system also maintains a constant yarn-to-rapier
angle, mitigating pick errors. The developed system offers a solution for expanding
weft selection in rapier looms, enabling more intricate fabric designs and increased
product versatility. To the best of our knowledge, this type of design has not been
tried before in weft selection.

## 1. Introduction

Textile industries produce consumer products that prioritize aesthetics, functional-
ity, or both. To cater to end-user preferences, the industry must diversify its offer-
ings. Introducing a variety of designs, colors, and materials is essential to meet
the demands of the current generation. For ready-made garments (RMG), fabric

**Data availability statement:** All relevant data are within the manuscript and its Supporting Information files.

**Funding:** The author(s) received no specific funding for this work.

**Competing interests:** The authors have declared that no competing interests exist.

serves as the primary raw material, and numerous manufacturing techniques and machines are utilized to enhance productivity or introduce a variety of products to improve aesthetics and texture. Rapier looms, which are commonly used in the textile industry for producing woven fabrics, are known for their reliability in manufacturing fabrics with wider widths. These looms incorporate advanced automation and innovative mechatronic solutions to enable seamless machine operation and efficient performance. Despite their high speed and significant role in balancing quality and productivity, the limited weft selection capacity of rapier looms restricts the ability to produce complex fabrics. This limitation is particularly evident in the restricted variety of weft yarns in terms of color, count, or material. Although warp yarns can feature greater variety through the use of sectional warping machines, the current design of weft selectors prevents the introduction of a broader range of weft yarns. Consequently, the diversity and complexity of fabric design are inherently limited. This study focuses on developing a new weft selector based on a mechatronic system to enhance the weft selection capacity, enabling the production of fabrics with complex designs, diverse color combinations, and intricate textures by utilizing multiple wefts.

The DORNIER P2 weaving machine features a proprietary and innovative color selection system, DORNIER DisCoS®. The Electronic Color Selector (ECS) uses filling selector needles to present the yarn for weaving. The P2 model provides a weft selection capability for up to 16 colors [1]. ITEMA offers a diverse portfolio of weaving machines, including models such as the R9500terry, R9500evo, and R9500evoterry. Though these weaving machines have many features, none have a weft patterning capacity of over 12 [2–4]. PICANOL, another leading manufacturer, offers various weaving machines tailored to pick insertion methods, product types, and more. In the rapier segment, PICANOL's offerings include the GTMax-i 3.0S Connect,and OptiMax-i Connect each with a filling selection capacity of up to 12 [5,6]. The "ONE" model in the GS series from SMIT integrates an EWS (Electronic Weft Selector) that allows weaving with 2, 4, 6, 8, or up to 12 colors. Similarly, the maximum weft selection capacity of the SMIT GS980F is capped at 12 [7,8]. At present, most rapier looms provide a maximum weft selection capacity of 16 colors. The weft selection capacity differs depending on the loom models and manufacturers. As a result, designers and manufacturers are constrained to create fabrics with a limited selection of weft yarns. An increased capacity in the weft selector enhances flexibility in manufacturing fabrics with intricate designs and diverse color variations. However, current technological limitations persist, requiring ongoing research to address these issues. Researchers are actively exploring various aspects of rapier looms to develop effective solutions to these challenges. For example, YANJUN XIAO and colleagues aimed to minimize energy consumption and operational costs by creating an integrated control system based on the direct drive of switched reluctance motors (SRM) [9]. Researchers have observed that stepper motors employed in selvedge devices and weft selection experience vibrations at lower speeds due to low-frequency oscillations, while at higher speeds, they generate back electromotive force (EMF). To address the first issue, subdivision control of motor winding

current has been implemented to stabilize the control system, and for the second, the authors propose a mixed current attenuation algorithm [10]. The problems associated with the warp tension fluctuation of the rapier loom and maintaining a constant warp let-off have been addressed in research. An advanced control system with fuzzy neural network and vector control has been implemented to enhance control on the warp tension and improve the drive performance of the let-off and take-up mechanisms [11]. Researchers have developed a new method for tension control in rapier looms to address issues such as low measurement accuracy, system complexity, and high cost. The method involves median and limiting filtering algorithms to eliminate disturbances and fluctuations in tension signals. Furthermore, the introduction of the number of learning times and the attenuation factor optimizes the performance of the backpropagation neural network [12]. In addition to enhancing the productivity of rapier looms, researchers are focusing on improving their versatility, particularly by increasing weft selection capacity. Soadbin Khan et al. introduced an Arduino-controlled filling yarn presenter powered by servo motors, using four colored yarns in practice, and proposed the inclusion of 32 weft yarn presenters. However, the study did not specify the angle at which the filling presenter delivers the weft yarn to the rapier. As the number of filling presenters increases, the angle between the weft yarn and the rapier changes. If the system offers weft at an unsuitable angle, it could prevent the rapier from effectively gripping the yarn from the filling presenter [13]. Egbers et al. designed a selection mechanism capable of choosing the desired weft from multiple weft yarns, with the process being programmatically controlled. A positioning system, governed by the program-controlled mechanism, moves the selected weft yarn into an active position by adjusting the weft yarn carrier. Once in this position, the weft yarn is ready for insertion into the weaving shed. However, the inventors did not disclose the maximum capacity of the selection device or how the yarn path might be influenced when a significant number of wefts are introduced [14]. Muller introduced a mechanism in which cam disks served as the primary controlling elements, their rotation driven by control pulls managed by a Jacquard system. This movement is transmitted to the rope via a roller follower and a sensing component. The position of the thread guide member, equipped with multiple eyelets for the weft threads, is likewise determined by this movement. This intricate setup positions the selected weft thread directly in front of the insertion apparatus. However, in today's era of automation, this design is unsuitable for integration with program-controlled modern looms, and its incorporation is impractical due to space limitations [15].

Therefore, opportunities remain to enhance the weft selection capacity. Increasing the number of wefts would open possibilities for creating intricate designs by incorporating a wider range of wefts. This study introduces a mechatronics approach to designing a prototype weft selector with enhanced weft selection capabilities. To realize this objective, a novel design was required. Adopting a mechatronic approach facilitates the implementation of the design in a practical operational setting. mechanical systems, control systems, electrical or electronic components, and information processing into a unified mechatronic system, indicating the inclusion of several disciplines such as electrical, software, and mechanical engineering. Mechatronic systems are widely used in modern automated machinery. For instance, researchers developed a pick-and-place robotic system for PCB assembly, robotic dispensing systems for hospital pharmacies, an automated apple sorting machine, and a single wheel balancing robot named GYROBO. These examples emphasize the reliability of mechatronic systems in achieving precision, effective control, and efficiency in versatile applications [16–20]. For this project, the mechanical components were fabricated locally, with many parts crafted from reused waste materials, while modular electronic driver boards were utilized to streamline the electronic system. Arduino's application in fields like home automation, healthcare, autonomous cleaning robots, educational robots, and pick-and-place robotic arm demonstrates its reliability in mechatronic control systems [21–24]. An Arduino Mega board was employed to execute tasks based on programmed instructions.

The primary objective of this research is to design and develop a prototype mechatronic weft selection system that enhances the weft section capacity of the rapier loom up to 20, exceeding the current limit of 16, to enable greater variation in weft color, count, and types. This system integrates mechanical and electrical components, as well as computer programming, using an Arduino-based control unit for precise and efficient weft selection.

 

The study introduces a novel circular arrangement of the weft feeder tubes, accommodating a large number of weft feeders, and employs a single weft selection module for selecting and presenting weft to the rapier. This advancement enhances the versatility of the rapier loom, allowing the production of multi-colored, textured, and complex fabric.

## 2. Mechanical design of the components

The automated weft selection system includes a control unit and mechanical components such as the Structural Frame, Weft Feeder Board, Weft Selection Module, and Prototype Rapier. The prototype's primary purpose is to exhibit the functionality of the newly developed selector.

### 2.1 Structural frame

The frame serves as the primary structure that supports all other components of the system, including the weft feeder board, weft selection module, and the prototype rapier. It is designed for modular adaptability, rigidity, and precise positional adjustment of the components while maintaining structural stability under dynamic loading during operation. Constructed from metal, the frame comprises four legs (aligned parallel to the Z-axis), each measuring 0.15 m in length. At the top, these legs are connected to a rectangular structure (parallel to the XY-plane) with dimensions of 0.15 m × 0.22 m (Fig 1a).

The top section of the frame is designed to accommodate the weft selection module and the control unit. It features two horizontally running metal plates (aligned parallel to the X-axis) extending the full length of the structure. These plates enable the weft selection module to slide back and forth, allowing its shaft to be positioned directly above the center of the circular cutout in the weft feeder board.

A mechanism is incorporated to securely hold the weft feeder board. This is achieved by using two opposing metal bars that support the board. These bars, attached to the legs of the frame (parallel to the Y-axis), are adjustable along the vertical (Z-axis) direction, allowing the weft feeder board to move accordingly. The board also has horizontal adjustability to ensure precise positioning. Key considerations for positioning the board involve ensuring that its center is vertically aligned with the shaft of the weft selection module and that the board lies in the same plane as the weft catcher.

The frame also includes a designated arrangement for holding the prototype rapier (aligned parallel to the X-axis). This configuration is similar to the arrangement designed for supporting the feeder board. Two additional metal bars (parallel to the Y-axis) are positioned parallel to those used for the weft feeder board, but are placed 0.305 m below them. This vertical distance is primarily determined by the length of the arm that delivers the weft to the prototype rapier.

### 2.2 Weft feeder board

The weft feeder board, depicted in Fig 1b, presents an innovative design in the context of weft selection mechanisms. Constructed from wood for maintaining a light-weight structure, the board is aligned in the XY-plane (shown in Fig 1a) and serves as a support structure for multiple weft feeder tubes. These tubes guide weft yarns with varying counts, colors, and material types. They are arranged in a circular pattern around a central hole with a diameter of 0.33 m, with uniform angular spacing determined by the total number of tubes used. This unique circular arrangement of the weft feeder tubes enables the weft selection module to reach and catch wefts from all the feeders The angular distance between the tubes is calculated as:

$$\alpha = 360°/n \tag{1}$$

Where $\alpha$ is the angular distance and $n$ is the number of feeder tubes

This prototype includes 20-weft feeders, resulting in an angular spacing of 18° between them. An increase in the number of weft feeder tubes correspondingly reduces the angular spacing. Each tube is oriented toward the center of the

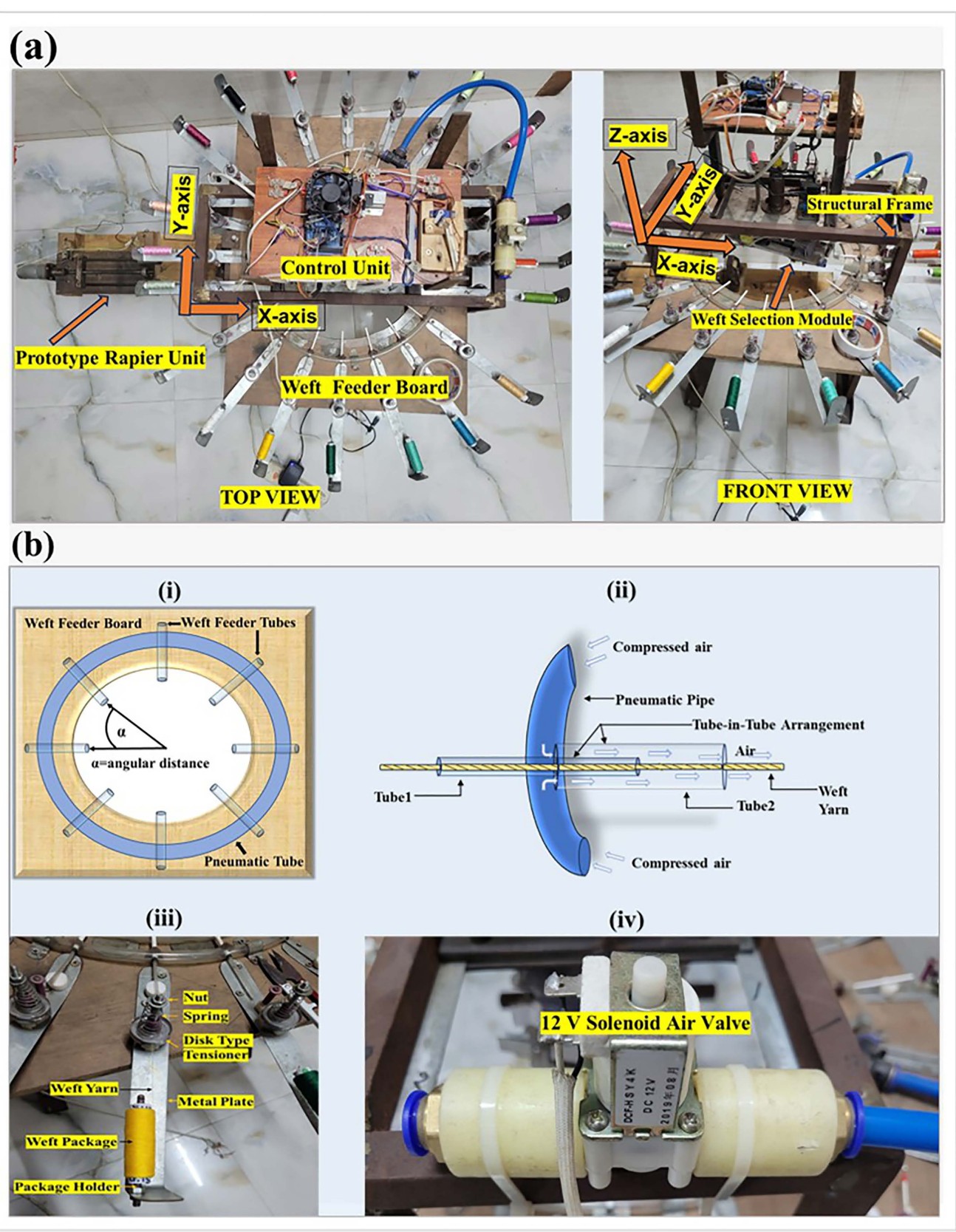

**Fig 1. (a) Top view and front view of the weft selection system with its different units, (b) Components of the weft feeder board: (i) the pneumatic tube and the weft feeder arrangement on the board, (ii) the tube-in-tube arrangement of the weft feeder, (iii) the package holder and disk tensioner, & (iv) the solenoid air valve.**

circular cut-out. A tube-in-tube structure is utilized to facilitate the passage of the weft yarn. The tube positioned closer to the tensioner, referred to as Tube1, has a smaller diameter than the tube closer to the weft catcher, referred to as Tube2. This arrangement creates a gap between the tubes, allowing for the passage of compressed air. Compressed air propels the weft yarn horizontally, ensuring its smooth capture by the weft catcher. To facilitate this process, the tubes are joined within a pneumatic pipe. Each weft feeder tube has the same structural design.

A pneumatic pipe encircles all tubes, forming a closed air delivery pathway to ensure equal air pressure to the feeder tubes. The airflow is regulated through a solenoid air valve connected to the pipe and controlled by a control unit. Airflow is activated only, when necessary, particularly when the weft catcher moves to grab the yarn. It provides some advantages, for instance: (i) reduces air consumption; (ii) reduces energy consumption; (iii) minimizes the chance of a change in yarn twist.

The board also accommodates the package holder and yarn tensioner, both of which are mounted on a metal plate secured to the board with screws. A disk-type tensioner is employed to provide necessary tension to the weft yarns, and its tension can be controlled by adjusting the spring using a nut, which can be turned clockwise or counterclockwise. Tension adjustment is important for avoiding looseness or tightness of the weft. This helps in the controlled unwinding of the yarn from the package.

## 2.3 Weft selector module

This is the main device for selecting and presenting weft yarn to the rapier carrier (Fig 2).

Typically, a separate weft selector is assigned to each yarn type, categorized by attributes such as color, count, or other characteristics. Consequently, the number of weft presenters or selectors must be at least equal to, or possibly greater than, the number of yarn varieties planned for inclusion in the fabric.

This design incorporates a single module capable of selecting a weft from diverse yarns and delivering it to the rapier. To achieve this, it must perform functions such as positioning for weft selection, capturing the correct weft, and delivering it to the rapier. The module consists of specialized components designed to perform these tasks effectively. This module is mounted on a metal plate that is engineered to slide horizontally along the metallic plate attached to the frame, allowing precise positioning. This alignment is critical as the shaft must correspond to the vertical axis (Z-axis) passing through the center of the weft feeder board. The clamps beneath the plate secure their positions once properly aligned.

A stepper motor is mounted on the plate to drive the rotation of the shaft connected to the weft selector. The motion from the stepper motor is transmitted to the shaft via timing pulleys and a belt. The weft selector is positioned by the shaft, which is driven by a VEXTA stepping motor, model number C7978-9012K-6. It ensures precise positioning of the module as it can rotate 0.9°/ step and holds the module in a steady position while the weft catcher is in action The weft selector primarily consists of two parts: the yarn catcher and the arm. The yarn catcher advances to retrieve the yarn and retracts while holding it. The yarn catcher is constructed from plastic to maintain lightweight and faster movement. It is customized to grip the yarn securely. It is mounted on a slider that moves within the grooves of the guide rails, enabling its forward and backward motion (illustrated in Fig 2i). These guide rails, crafted from repurposed plastic waste from LED light packaging, facilitate a smooth operation.

The movement of the slider is facilitated by a belt and pulley system. One end of the slider is connected to the front of the belt, whereas the opposite end is attached to the rear of the belt. Clockwise and counterclockwise rotations of the pulley, driven by two DC motors, enable the belt to move the slider back and forth.

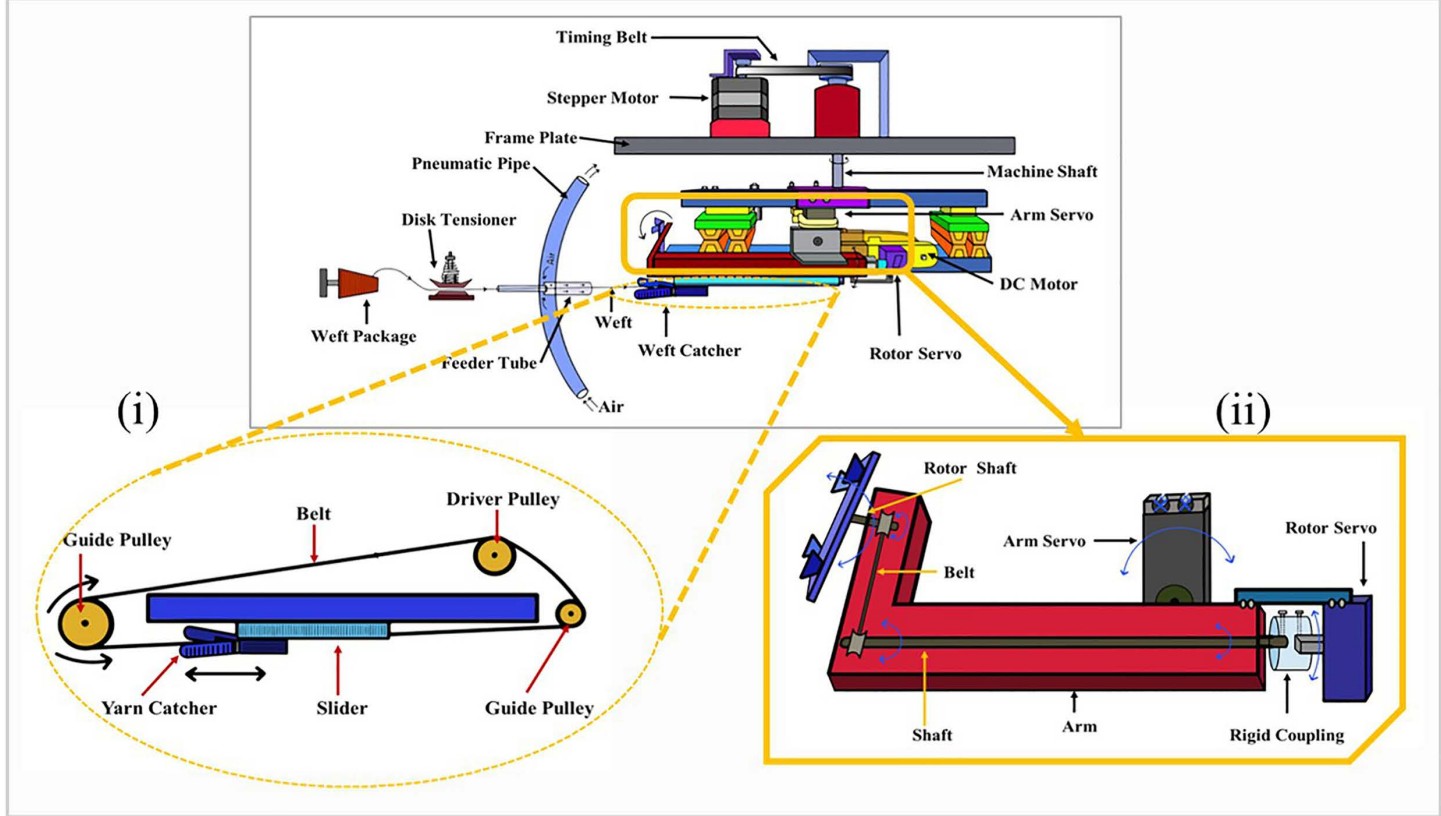

**Fig 2. The weft selection module: (i) the yarn catcher and the slider movement, (ii)the arm and the rotor.**

The arm and the rotor shown in Fig 2(ii) ensure that the yarn caught by the yarn catcher is presented at a 90-degree angle to the prototype rapier. To position the selected weft yarn precisely at the rapier, the arm performs a 100-degree angular motion driven by a servo motor. Because the relative positions of the weft selector and prototype rapier vary, the angle between the weft yarn and the rapier also fluctuates. This challenge is addressed by an innovative arm design that incorporates a rotor, ensuring the weft is consistently presented at a 90-degree angle to the rapier. The rotor, connected to a rotor shaft, receives motion via a belt that links the arm shaft to the rotor servo through a coupling mechanism. In this manner, the rotor servo transmits motion to the rotor. An intelligent computer program synchronizes the rotor position depending on the module position.

## 2.4 Prototype rapier

The prototype rapier (Fig 3) demonstrates how the automated weft selection system works.

This prototype is categorized as a single rigid rapier, designed to simplify the mechanical interface while preserving the essential kinematic characteristics of industrial rapier looms. s It is constructed using a rod and iron sheet shaped into a rapier gripper. The gripper consists of two jaws—an upper and a lower jaw—that remain closed. The lower jaw features a bow-shaped curve, whereas the upper jaw has a straight design at the opposite end. A spring is placed in between, which exerts pressure so that the front part of the jaws remains closed in normal conditions.

The movement of the rapier is facilitated by a bolt-screw mechanism. The rapier rod is attached at one end to a solid metal bar, whereas the other end features the gripper. Two rods are connected to the metal bar: a rapier rod and a

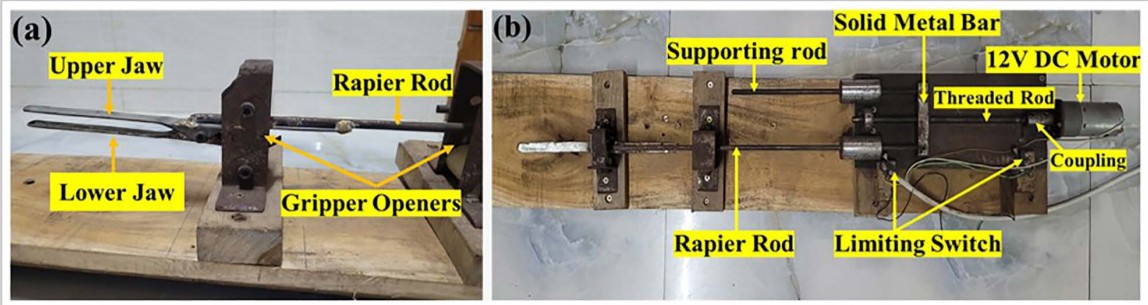

**Fig 3. (a) Rapier head consisting of upper jaw and lower jaw, (b) top view of the prototype rapier arrangement.**

supporting rod. Linear motion is achieved through the metal bar, with a threaded rod serving as the bolt. This mechanism is enabled by threading the rod entirely and machining a matching threaded hole at the center of the metal bar. The rotation of the bolt causes the metal bar to move linearly, along with the rapier and the supporting rod. The supporting rod prevents the metal bar from rotating during the rotation of the bolt. The bolt is coupled to a DC motor shaft at one end, whereas the other end is mounted on a bearing to facilitate smooth rotation.

The gripper jaws are opened using two gripper openers—one located at the forwardmost position where the rapier collects the weft yarn from the weft selector arm, and the other at the backwardmost position where the rapier completes the weft insertion and releases the yarn. Each gripper opener is designed simply, consisting of two rollers with a gap between them. As the bow-shaped curvature of the rapier enters the gap, it undergoes compression because the gap is narrower than the height of the bow, causing the front end of the jaws to open. The rapier stroke limits are regulated by two limit switches.

## 3. Control unit development

The circuit diagram of the Automated Weft Selection System is shown in Fig 4.

The system architecture incorporates essential components, including an Arduino Mega, an L293D motor driver shield for controlling two DC motors, two servo motors, one stepper motor, an additional L293D driver for a 12V DC motor operating the prototype rapier, limit switches for position regulation, a 12V solenoid air control valve, a servo motor, and a normally-open held-closed switch for managing airflow through pneumatic pipes. A 6V and a 12V DC adapter provide power to the components.

A circuit was designed and simulated in TINKERCAD to observe the movement of the arm servo (S1) and rotor servo (S2). It is provided as a supplementary document (S1 Appendix).

The Arduino Mega microcontroller board was selected for this project due to its higher number of I/O pins (16 analog, 54 digital, and 15 PWM outputs) and larger flash memory, which makes it ideal for handling complex tasks. The Arduino Mega is well-suited for this application, equipped with an ATmega2560 processor operating at $1.6 \times 10^7$ Hz [21]. The board was programmed using the Arduino IDE to perform the specific functions required by the Automated Weft Selection System.

The L293D Motor Driver Shield is employed to manage various motors integrated into the system via Arduino. Positioned atop the Arduino board, the shield connects through its male headers to the female headers on the board. The shield is designed to control the DC motors, stepper motors, and servo motors. It incorporates a 74HC595 shift register and two L293D motor driver ICs. Each L293D integrates two H-bridges, enabling control of two DC motors or one stepper motor simultaneously, with a current capacity of 0.6 A per channel. Accordingly, the M1 and M2 ports connect to two DC motors, while the M3-M4 port is assigned to the stepper motor.

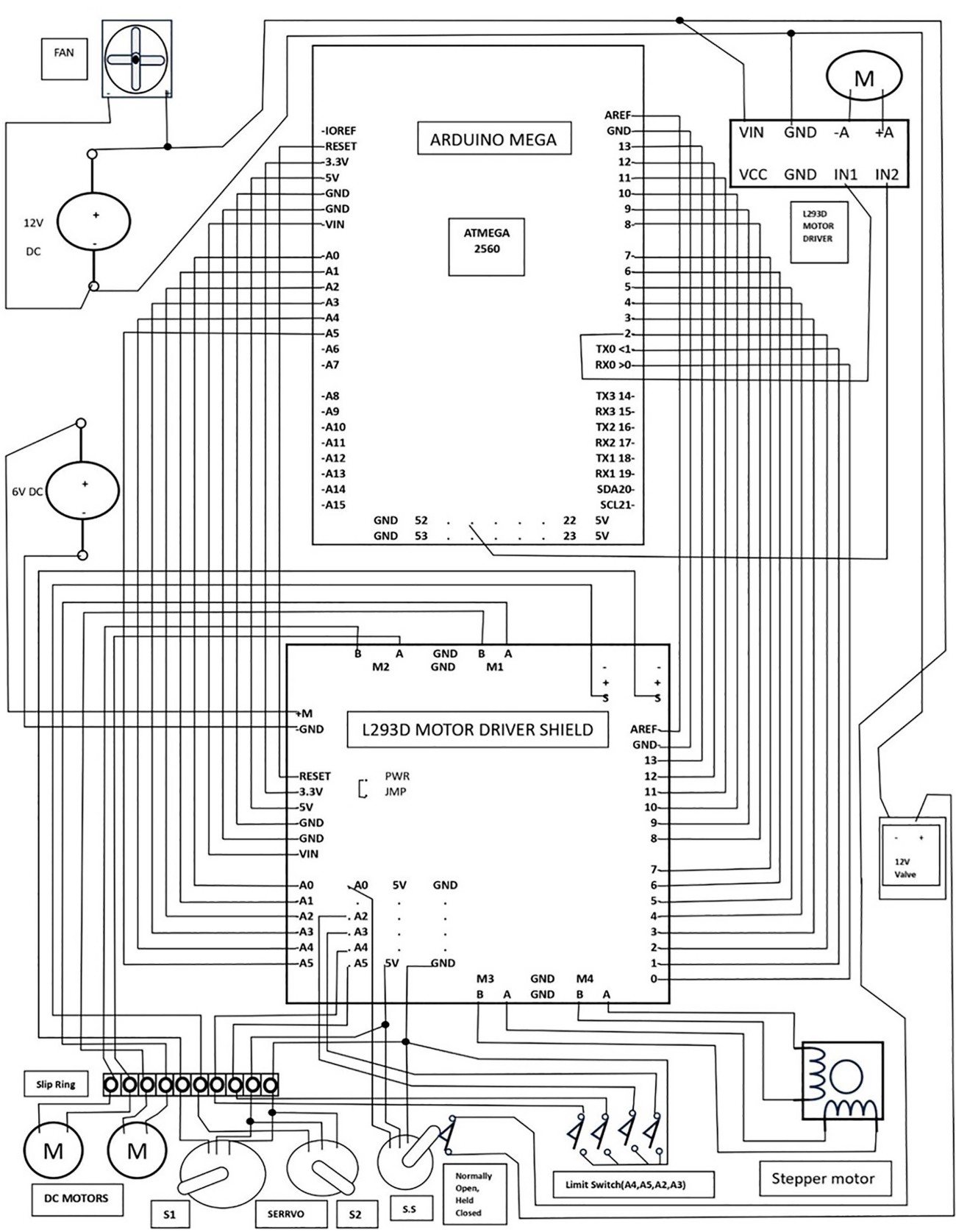

**Fig 4. Schematic of the circuit design of the prototype weft selection device.**

The shield features two 3-pin servo headers, each comprising a signal pin, a 5V pin, and a ground pin. Signal wires (orange) from Servo1 (S1: arm servo) and Servo2 (S2: rotor servo) are linked to the signal pins. Servo1 (S1) and Servo2 (S2) interface with the Arduino's digital pins D10 and D9, respectively. D3 to D8 and D11 to D12, these eight pins are used to control the stepper motor and DC motors. The shield does not utilize pins D2 and D13, leaving them available for future use. Additionally, the shield interfaces with the Arduino board's analog pins (A0-A5), ensuring these pins remain accessible via the shield. Pins A2-A5 are allocated to connect the limit switches.

Two limit switches are employed to define the slider's forward and backward positions, with one terminal of each switch linked to pins A4 and A5, and the other terminal grounded. Similarly, pins A2 and A3 are connected to the limit switches positioned at the rapier's extreme forward and backward points, with the other terminals of these switches grounded. Pin A5 functions as the signal pin for the switch servo (S.S). The shield provides 5V and ground rails for power distribution. All the servo motors are powered by connecting their 5V and ground wires to the common 5V and ground rails provided by the shield. The switch servo operates a push-button switch, essentially a normally open, held-closed limit switch, which is responsible for controlling the 12V solenoid air valve. A relay was avoided in this design to prevent mechanical wear and contact welding caused by the high voltage. The EXT_PWR terminal is supplied with a 6V power source to operate the electrical components connected to the shield. Additionally, the power supply jumper, marked as PWR, is used to deliver power to the Arduino board. A separate L293D controller is employed to manage a 12V DC motor that drives the prototype rapier. The motor interfaces with terminal A (A+, A-). Two LEDs associated with this terminal blink during the forward and reverse movements of the motor. The IN1 and IN2 pins of the controller are linked to Arduino pins 2 and 44, which support PWM signaling. The motor is powered through the GND and VIN terminals, supplied with 12V via an adapter. The airflow in the pneumatic pipe is regulated by a 12V solenoid air valve. The valve is powered by a 12V supply through a normally open, held-closed switch controlled by a servo. The angular motion of the servo horn activates the switch and completes the circuit. A 12V cooling fan is installed to dissipate heat, particularly from the motor driver ICs.

The system consists of several electrical and mechanical components. Although attempts were made to ensure precision and maintain quality during manufacturing, the machine's overall durability depends mostly on the reliability and effectiveness of these components.

### 3.1 Electrical performance analysis

During the operation of the weft selection system, the electrical performance of the system was evaluated to determine its overall power requirement and energy consumption. The device operates in two different supply voltages. A maximum voltage of 5.5 V ($V_1$) was recorded with a corresponding current drawing of 1.202 A ($I_1$), and another was 12 V ($V_2$) and 0.6 A ($I_2$). The test was conducted for a duration of 60 seconds. The power requirements can be calculated as:

$$P = V_1\, I_1 + V_2\, I_2$$

$$= (5.5 \times 1.202) + (12 \times 0.6)$$

$$= 13.81\ \text{W}$$

And the energy consumption of the system is:

$$E = P \times t$$

$$= 13.81 \times 60$$

$$= 828.6 \text{ J}$$

$$\approx 0.23 \text{ Wh}$$

## 4. Operational methodology

The uniqueness of the newly developed automated weft selection system lies in its ability to select all the necessary wefts and manage the circular arrangement of the weft yarns. The system operates in four distinct steps, as shown in Fig 5. The free movement of the weft selection module, demonstrating the workability of the components, is presented in the supporting video (S1 Video).

**Step 1 (Positioning):** The necessary wefts are organized in a circular layout. The system can accommodate up to 20 distinct yarns. The module positions itself in front of the appropriate weft feeder to select the desired yarn based on the weft pattern of the fabric. Suspended from the shaft, the module is actuated by a stepper motor via a timing belt, which is depicted in Fig 5a. Table 1 represents the necessary parameters for the module's movement.

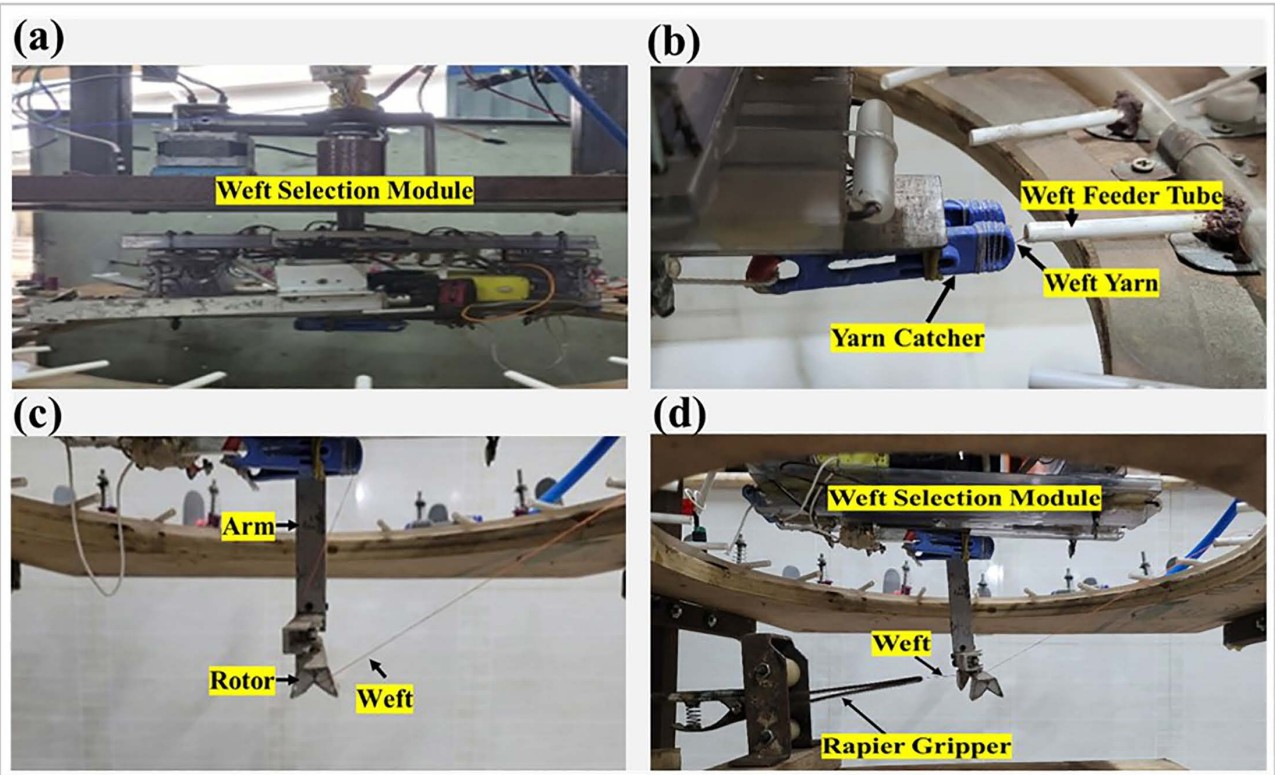

**Fig 5. Different steps involved in weft selection and insertion: (a) Positioning, (b) Yarn catching, (c) Yarn taking down and placement, (d) Yarn gripping.**

**Table 1. Symbols, parameters, formulas, and the values involved in the weft selection module movement.**

| Symbol | Parameter | Formula | Value |
|---|---|---|---|
| $\theta$ | Total angle | | 360º |
| $n$ | Total number of weft feeders | | 20 |
| $\alpha$ | Rotation of the machine(module) shaft (driven) to reach the adjacent feeder | $\alpha = \theta/n$ | 18º |
| $T_1$ | Driver teeth | | 20 |
| $T_2$ | Driven teeth | | 40 |
| $\beta$ | Rotation of the motor shaft (driver) to drive the module shaft (driven) to reach the adjacent feeder | $\beta = \alpha \times \frac{T2}{T1}$ | 36º |
| $\gamma$ | Rotation of the motor shaft in degrees per step | | 0.9º |
| $s_1$ | Number of steps made by the driven shaft to reach the adjacent feeder | $s = \beta/\gamma$ | 40 |

Consequently, the stepper motor completes 40 steps to position the module with the subsequent weft feeder (Table 1). In cases where a feeder is inactive, such as weft-feeder2, the motor executes 80 steps to move directly from weft-feeder1 to weft-feeder3.

**Step 2 (Yarn Catching):** Once the weft selection module aligns with the correct weft feeder tube, the weft catcher advances to capture the weft. The weft catcher is mounted on a slider that moves along two guide rails. A rope and pulley mechanism is employed to shift the slider along the guide rails, with a pulley linking the shafts of two 5V DC motors. When the pulley rotates in one direction, it moves the slider forward, and when the motors reverse the rotation, the slider is pulled in the opposite direction. This mechanism enables the slider to travel both forward and backward. Once the slider, with the yarn catcher, reaches the forward position (Fig 5b), it activates a limit switch (A4), sending a signal to the Arduino. According to the algorithm, this halts the motor movement and opens the 12V solenoid air valve to allow airflow through the pneumatic pipe and weft feeder tubes. The air pressure differs for different types of yarn, as shown in Table 2. The air stream causes the weft yarn to float horizontally, aiding the yarn catcher in capturing it. Upon capturing the yarn, the yarn catcher immediately begins to move backward as the motor shafts rotate in the reverse direction. The air pressure is adjusted based on the yarn used.

Limit switch A5 detects when the slider has reached its rearmost position. Subsequently, the slider moves slightly forward to disengage from the limit switch, initiating the yarn-taking operation.

**Step 3 (Yarn taking down and placement):** The arm and rotor, controlled by two servo motors, work together to accurately lower and position the weft yarn (Fig 5c). The arm of the weft selector module descends, delivering the selected yarn to the rapier. A continuous servo motor (arm servo, MG995) drives the arm, rotating it from 100° at its highest position to 0° at its lowest position, where it feeds the rapier. The rotor positions the weft yarn at a 90° angle relative to the rapier (shown in Fig 6), operated by a TowerPro MG90S servo motor (rotor servo).

**Table 2. List of the required pressure for different yarn counts.**

| Yarn Count | Air Pressure (Kg/cm²) | Air Pressure (Pa) |
|---|---|---|
| 30 Ne (viscose) | 0.5 | $4.903 \times 10^4$ |
| 40 Ne (viscose) | 0.5 | $4.903 \times 10^4$ |
| 40 Ne (spun polyester) | 0.5 | $4.903 \times 10^4$ |
| 40 Ne (Carded yarn) | 0.6 | $5.88 \times 10^4$ |
| 6 Ne (Open end) | 0.7 | $6.86 \times 10^4$ |
| 20 Ne (Open end) | 0.55-0.6 | $5.39 \times 10^4$-$5.88 \times 10^4$ |
| Swing thread | 0.58 | $5.69 \times 10^4$ |
| Polyester 50 Denier of 96 filament & Lycra 40 denier | 1.3 | $1.27 \times 10^5$ |

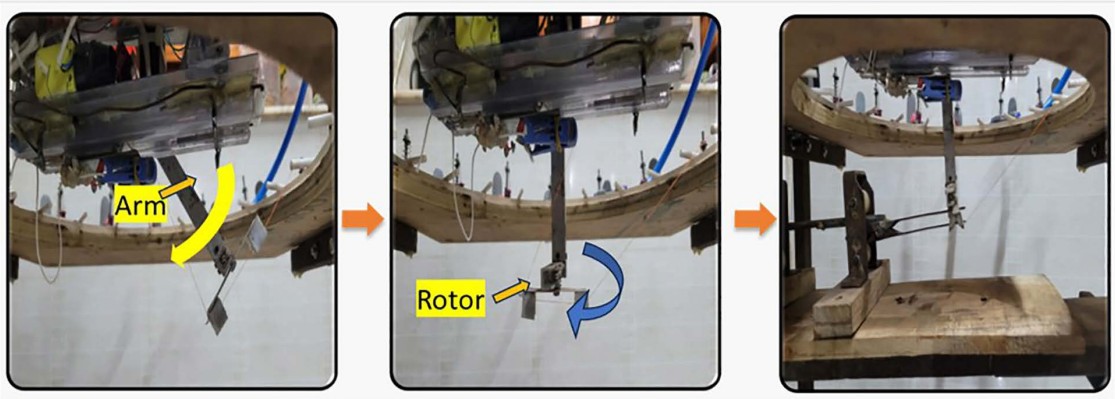

**Fig 6. Movement of the arm and the rotor.**

The Arduino determines the number of steps taken by the stepper and the degree of movement of the selection module from its starting point. It calculates the position of the selection module and signals the rotor servo to align the rotor at a 90° angle to the rapier. When the arm is at its highest point (100°), the rotor makes a 90° angle, parallel to the Z-axis. At its lowest point (0°), the rotor adjusts to ensure the yarn is positioned at a right angle to the rapier. Steps 1–3 are shown in the supporting file, S2 Video.

**Step 4 (Yarn Gripping):** When the yarn is successfully taken down and positioned correctly, the prototype rapier begins its movement. It advances to grip the weft where the gripper opener opens its jaws. The last three steps [2–4] are shown in the supporting video (S3 Video).

After securing the yarn (Fig 7), the rapier begins to move backward (S4 Video) to complete the weft insertion. The limit switches signal the Arduino about the rapier's position, enabling the Arduino to control the rapier's driving motor accordingly.

These four steps—positioning, yarn catching, yarn taking down and placement, and weft insertion—comprise the complete weft selection and insertion process. Steps 1–3 are focused on weft selection, while step 4 pertains to weft insertion; the sequences are shown in Fig 8. However, the sequence of these steps will be according to the weft insertion pattern.

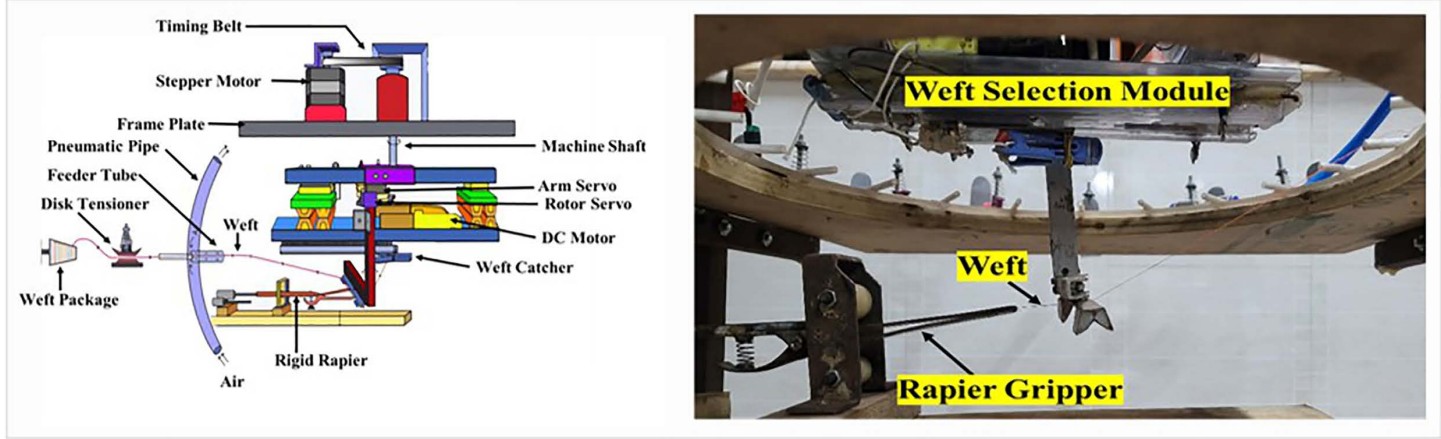

**Fig 7. Yarn gripping by the rapier selected by the weft selector (in the left illustration of the weft selection device and on the right, the device in functioning condition).**

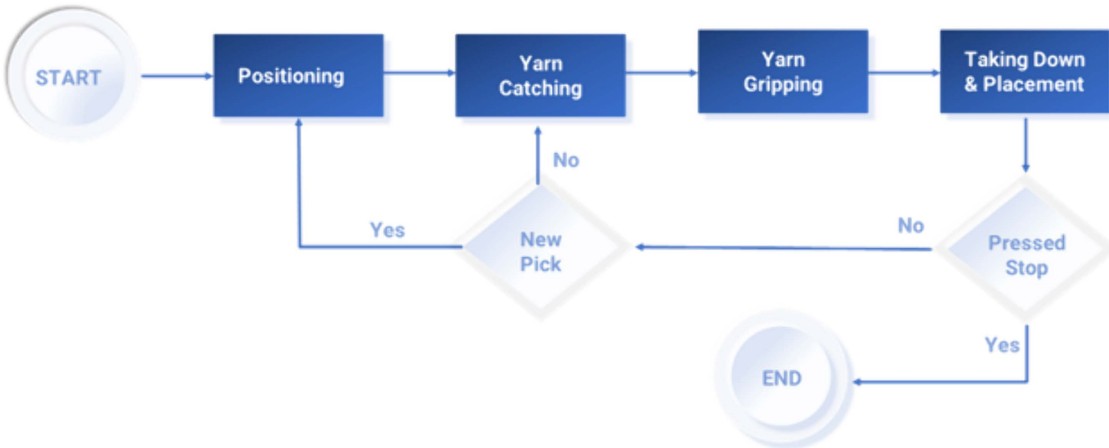

**Fig 8. The sequence of the steps in weft selection.**

For instance, a weft pattern of 3, 2, 0, 1, 1, 1, 1, 1, 1, 1, 1, 1, 1, 1, 1, 1, 1, 1, 1, 1 may be considered. This implies that yarn from feeders 1 and 2 needs to be inserted three and two times, respectively, whereas no yarn is taken from feeder 3. Yarn from feeders 4–20 is collected once each.

To select and insert the weft three times from feeder 1, the weft selection device will perform the 1st to 4th steps once, and the 2nd to 4th steps for the next two times; after that, the next two wefts will be taken from feeder 2. To make this happen, the device will perform the 1st to 4th steps once, and then next time, it will perform the 2nd to 4th steps.

As this weft pattern does not have any weft from feeder 3, the weft selection device will perform the 1st step (positioning), this time in front of feeder 4, skipping feeder 3, and then the 2nd to 3rd steps for one time.

The actuation time was determined using different numbers of wefts, for instance, 20, 15, 10, and 5 wefts. The actuation time for the corresponding weft patterns has been summarized in Table 3.

From the data presented in the above-mentioned table, the picks per minute (PPM) were determined to be 9.

## 5. Discussion and implications

To evaluate the performance of the weft selection module, various yarn types were used to assess its selection capabilities. Every yarn type was fed into all 20 feeders, and 5 trials were conducted for each yarn Thenumber of successful selections was recorded. The test was carried out in a woven fabric manufacturing facility using the available weft yarns (11 different yarns) in production at the time, including 30Ne Viscose, 40 Ne Viscose, 40 Ne Spun Polyester, 40 Ne Carded Yarn, 6 Ne Open End, 20 Ne Open End, Polyester 50 Denier with 96 filaments, and Lycra 40 Denier, along with sewing thread. The test yielded remarkable results; the average weft selection rate and pick insertion rate were 99.36%, and 99.27% which is close to a 100% success rate. After the test miss-pick rate was recorded as 0.73% (see S2 Appendix).

**Table 3. Actuation time for different numbers of wefts.**

| Number of wefts | Weft pattern | Actuation time in seconds per weft | PPM (picks per minute) |
|---|---|---|---|
| 20 | (1,1,1,1,1,1,1,1,1,1,1,1,1,1,1,1,1,1,1,1) | 6.66 | 9 |
| 15 | (1,1,1,1,1,1,1,1,1,1,1,1,1,1,1) | 6.65 | 9 |
| 10 | (1,1,1,1,1,1,1,1,1,1) | 6.65 | 9 |
| 5 | (1,1,1,1,1) | 6.62 | 9 |

To meet the research goals, a novel design was developed, featuring several remarkable attributes, illustrated in Fig 9; primarily, the prototype ensures a fixed angle between the weft yarns and the rapier. A component of the prototype, called the rotor, positions the weft at a 90° angle in front of the rapier and is controlled by the Arduino microcontroller through a complex program. This minimizes the chance of miss-picking, as the angle remains constant despite an increase in the number of weft yarns.

Secondly, this design enhances the weft yarn selection capacity, currently set at 20, though this number is not fixed. The capacity can be further increased by reducing the spacing between the weft feeder tubes.

Thirdly, the weft yarns are presented via circularly arranged weft feeder tubes. Finally, only a single selector is employed to choose the desired yarn from any of the 20 feeder tubes. The weft selection and the weft pattern can be programmed by the user. The device is programmed to accommodate any weft pattern provided by the user. If the desired weft in the pattern is not from a particular tube, the device will bypass that weft feeder tube.

Furthermore, the proposed weft selection system exhibits high scalability and potential for industrial deployment. The modular design of the weft selector and the circular arrangements of the feeders allow the selection capacity to be expanded beyond the current 20, enabling adaptation of a wide range of wefts and color combinations. The reconfiguration of the control algorithm and use of the encoder with the loom's main shaft facilitate synchronization with high-speed commercial looms, facilitating seamless integration into existing production machines. With appropriate actuator optimization and material selection, the system can be developed into an industrial-grade unit capable of continuous operation under typical weaving conditions.

## 6. Conclusion

The fundamental methods of weft selection have remained largely unchanged, limiting the variety of available wefts. This research introduces a mechatronic solution to overcome the limitations of existing weft selectors with smaller selection capacities by significantly expanding the weft selection range, enabling the inclusion of more diverse wefts in terms of count, color, and material. The Arduino platform is leveraged for designing the system, which allows flexibility and scalability in selecting weft yarns. The weft arrangement system is designed to accommodate up to 20 weft yarns, though this number

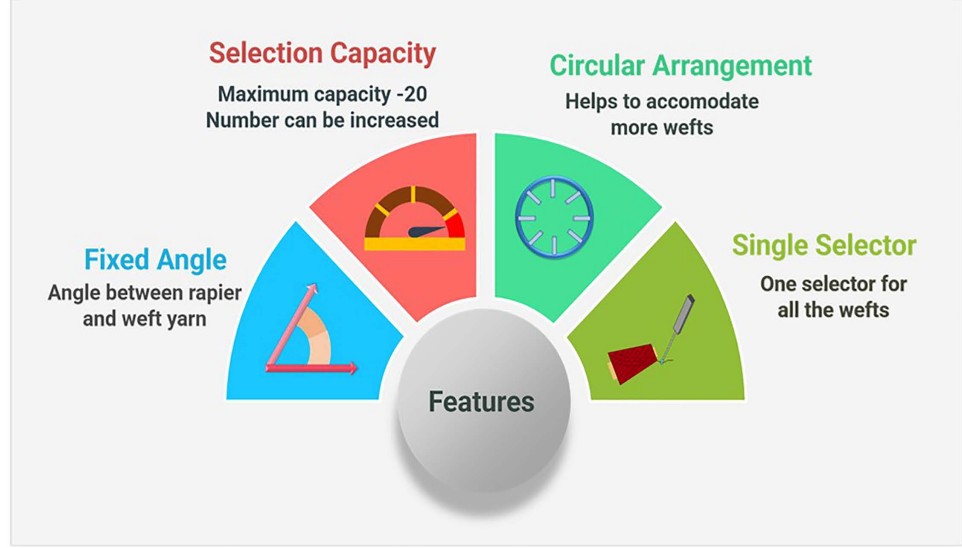

**Fig 9. Features of the developed weft selection device.**

can be increased by reducing the spacing between the feeder tubes. Instead of using multiple selectors for different yarn colors, this system uses a single selector module capable of choosing all required weft yarns. The prototype demonstrates the functionality and increased selection capacity. The inclusion of a rotor to maintain a constant yarn-to-rapier angle further reduces the risk of picking errors, ensuring consistent performance. The device successfully operated with a wide range of yarn at a speed of 9 PPM with the average weft selection, pick insertion, and miss-pick rate of 99.36%, 99.27% and 0.73% respectively, validating the system's practical potential to handle a variety of yarn effectively. The power requirement of the system was 13.81 W, and the energy consumption was recorded as 826 J (≈ 0.23 Wh). As a prototype version, the system's power requirement is less, which is also reflected in its speed. The next stages will focus on improving synchronization with loom cycles and integrating the system into industrial looms. This involves installing an encoder on the main shaft to record real-time loom data, allowing for accurate component position analysis and streamlining the weft selection module's functioning via a synchronization algorithm, to support large-scale production. The developed system enhances versatility in fabric production, facilitating more complex and varied designs while improving overall functionality.

## Supporting information

**S1 Video. Free movement of the weft selection module.**
(MP4)

**S2 Video. Demonstration of steps 1–3 (Positioning, Yarn Catching, Yarn taking down, and placement) of the weft selection module.**
(MP4)

**S3 Video. Demonstration of steps 2–4 (Yarn Catching, Yarn taking down and placement, Yarn Gripping).**
(MP4)

**S4 Video. Yarn gripping by the prototype rapier.**
(MP4)

**S1 Appendix. TINKERCAD circuit diagram.**
(DOCX)

**S2 Appendix. Weft selection and insertion rate.**
(DOCX)

## Acknowledgments

The authors gratefully acknowledge the support of NOMAN TEXTILE MILLS LTD.(NTML). Thanks to Mohammad Abdullah Zaber for allowing us to carry out the project in NTML. We are also grateful to Jahidul Hoque Majumder, Md. Aminul Hoque and Akram Hossain for their help.

## Author contributions

**Conceptualization:** Aiead Ibne Fahim, Md. Mohaddesh Hosen.

**Data curation:** Aiead Ibne Fahim.

**Formal analysis:** Aiead Ibne Fahim.

**Investigation:** Md. Mohaddesh Hosen.

**Methodology:** Md. Abdullah Al. Mamun.

**Resources:** Shah Alimuzzaman Belal.

**Software:** Aiead Ibne Fahim.

**Supervision:** Md. Mohaddesh Hosen, Md. Abdullah Al. Mamun, Shah Alimuzzaman Belal.

**Visualization:** Aiead Ibne Fahim, Md. Abdullah Al. Mamun.

**Writing – original draft:** Aiead Ibne Fahim.

**Writing – review & editing:** Md. Mohaddesh Hosen, Md. Abdullah Al. Mamun.

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
