## [Decision Letter · Decision Letter 0]

3 Sep 2025

PONE-D-25-37019Towards Devising a Novel Mechatronic Weft Selection Module for Enhancing Weft Selection Capacity in Rapier LoomsPLOS ONE

Dear Dr. Al. Mamun,

Thank you for submitting your manuscript to PLOS ONE. After careful consideration, we feel that it has merit but does not fully meet PLOS ONE’s publication criteria as it currently stands. Therefore, we invite you to submit a revised version of the manuscript that addresses the points raised during the review process.

We look forward to receiving your revised manuscript.

Kind regards,

Ying Ma, Ph.D.

Academic Editor

PLOS ONE

Journal Requirements:

2. We note that your Data Availability Statement is currently as follows: All relevant data are within the manuscript and in Supporting Information files.

Reviewers' comments:

Reviewer's Responses to Questions

**Comments to the Author**

1. Is the manuscript technically sound, and do the data support the conclusions?

Reviewer #1: Partly

Reviewer #2: Yes

2. Has the statistical analysis been performed appropriately and rigorously? 

Reviewer #1: Yes

Reviewer #2: Yes

3. Have the authors made all data underlying the findings in their manuscript fully available?

Reviewer #1: Yes

Reviewer #2: Yes

4. Is the manuscript presented in an intelligible fashion and written in standard English?

Reviewer #1: No

Reviewer #2: Yes

5. Review Comments to the Author

Reviewer #1: Review Comments to the Author

This manuscript presents the design, development, and demonstration of a novel mechatronic weft selection module for rapier looms. The authors propose a circular feeder-based programmable selection system capable of handling up to 20 weft yarns, supported by an Arduino Mega-based controller and mechanical automation. The approach is innovative and practically relevant, particularly in addressing the long-standing limitations in weft pattern diversity on rapier looms.

Strengths of the Study:

• The manuscript addresses a significant technological limitation in woven fabric production and offers a novel hardware solution grounded in real-world constraints.

• The integration of mechanical, pneumatic, and electronic components is well thought out and economically viable, especially the use of low-cost controllers and reused materials.

• A full prototype is developed and tested with various yarn types, achieving a 100% selection and insertion success rate.

• The paper is generally well-written and structured with clear intent and progression.

Areas for Improvement:

1. Title:

• The current title is informal and includes redundant phrasing. Suggested revision:

"Development of a Mechatronic Weft Selector to Enhance Patterning Capacity in Rapier Looms"

2. Abstract:

• Add quantitative results (e.g., number of trials, speed, types of yarns).

• Avoid vague novelty claims unless supported by comparative literature.

3. Introduction:

• The literature review should include more peer-reviewed technical papers rather than brochures.

• Clearly state the research objectives and novel contributions at the end of the section.

4. Design and Methods:

• Improve scientific framing of design descriptions (currently too procedural).

• Embed all referenced figures.

• Add performance metrics (actuation timing, power consumption, durability).

• Consider including CAD diagrams or exploded views of the module.

5. Operational Methodology:

• Add visuals such as flowcharts or timing diagrams.

• Move TINKERCAD simulation details to supplementary materials.

6. Results and Evaluation:

• Benchmark your system against existing commercial devices (DORNIER, ITEMA, PICANOL).

• Include more performance metrics: insertion time, miss-pick rate, energy use.

7. Discussion:

• Excellent recognition of the rotor's role in angular stabilization.

• Add commentary on system scalability and industrial deployment potential.

8. Conclusion:

• Reiterate key performance outcomes.

• Clarify next steps in industrial integration and synchronization with real loom cycles.

9. References:

• Include more peer-reviewed references to increase academic credibility.

Suggested Additional References for Future Directions:

1. Azizan, A., Cao, S., Dahlan, A., & Endrini, S. (2025).

Mapping knowledge landscapes and emerging trends in digital biomarkers for dementia in older adults: A scoping and bibliometric analysis.

Archives of Gerontology and Geriatrics Plus, 2(2), 100148.

https://doi.org/10.1016/j.aggp.2025.100148

Why include it? Digital biomarker integration can inspire future smart textile applications or sensor-augmented fabrics.

2. Azizan, A., Endrini, S., & Abdullah, K. H. (2025).

A research landscape analysis on Alzheimer’s disease and gerontechnology: Identifying key contributors, hotspots, and emerging trends.

Archives of Gerontology and Geriatrics Plus, 2(1), 100125.

https://doi.org/10.1016/j.aggp.2025.100125

Why include it? Gerontechnology overlaps with future applications of adaptive smart textiles for healthcare and elderly monitoring.

3. Azizan, A., Sirada, A., & Samosir, N. R. (2025).

Falling Forward: Tracing Technological Solutions for Fall Prevention in Older Adults (1996–2024).

Information Research Communications, 1(2), 74–82.

Why include it? Presents technological pathways (e.g., sensor-based prevention) that can inspire smart fabric integration in advanced looms.

Final Recommendation:

This manuscript proposes a highly novel and practically valuable weft selection mechanism that enhances design flexibility in rapier looms. However, before acceptance, the authors should revise the manuscript to address the outlined technical and structural concerns. With these improvements, the paper will make a significant contribution to the fields of textile engineering and industrial automation.

Reviewer #2: This manuscript proposes a mechatronic weft selection system for rapier looms, targeting up to 20 yarns with a fixed yarn-to-rapier angle. The concept is novel and relevant, but the technical validation, integration details, and industrial feasibility are incomplete.

(1)How was the claimed 100% selection success verified over long production runs?

(2)Is there data on operational speed and any effect on loom throughput?

(3)Were yarn breakages or mis-picks quantified across yarn types?

(4)How stable is the angle control when feeder spacing is further reduced?

(5)Why were no industrial weft selection benchmarks used for comparison?

(6)Could pneumatic air pressure variation impact yarn quality over time?

(7)How is the Arduino control synchronized with actual loom cycles?

(8)Are there durability tests showing wearing of moving parts and guides?

(9)What is the cost implication versus existing multi-selector systems?

(10)Could vibration or electrical noise affect controller accuracy in real looms?

6. PLOS authors have the option to publish the peer review history of their article (what does this mean? ). If published, this will include your full peer review and any attached files.

**Do you want your identity to be public for this peer review?** For information about this choice, including consent withdrawal, please see our Privacy Policy .

Reviewer #1: No

Reviewer #2: No

---

## [Author Response · Author response to Decision Letter 1]

7 Nov 2025

The file containing the responses has been uploaded to the submission portal.

Response to the Reviewers

The authors are grateful to the reviewers for their valuable comments. The manuscript has been revised in accordance with the provided advice. The correction has been highlighted in red in the revised version of the manuscript with track changes.

Reviewer 1:

1. Title:

• The current title is informal and includes redundant phrasing. Suggested revision:

"Development of a Mechatronic Weft Selector to Enhance Patterning Capacity in Rapier Looms"

Author’s reply: Necessary correction has been made according to the suggestion.

2. Abstract:

• Add quantitative results (e.g., number of trials, speed, types of yarns).

Author’s reply: The above-mentioned quantitative results have been added to the Abstract, and also a supplementary document titled “Weft Selection and Insertion Rate” (S2 Appendix) has been included, providing detailed information on the yarn types used and the corresponding number of trials conducted.

• Avoid vague novelty claims unless supported by comparative literature.

Author’s reply: Owing to the commercial nature of loom manufacturing, limited data are available in the form of published research articles. Therefore, relevant research papers, patents, and technical brochures were thoroughly reviewed to identify existing weft selection capacities, which, to the best of our knowledge, reach up to 16. Based on this analysis, the proposed system introduces a redesigned weft selection mechanism that extends this capacity to 20, thereby demonstrating a measurable improvement over the current state of the art.

3. Introduction:

• The literature review should include more peer-reviewed technical papers rather than brochures.

Author’s reply: Necessary correction has been made according to the suggestion. Brochures with lower weft selection capacity have been removed (Previous reference no. 3, 4, 5, 7, 10, 12, 13), and new research articles have been added.

• Clearly state the research objectives and novel contributions at the end of the section.

Author’s reply: Necessary correction has been made according to the suggestion.

4. Design and Methods:

• Improve scientific framing of design descriptions (currently too procedural).

Author’s reply: Necessary correction has been made according to the suggestion.

• Embed all referenced figures.

Author’s reply: Necessary correction has been made according to the suggestion.

• Add performance metrics (actuation timing, power consumption, durability).

Author’s reply: Information regarding durability, power consumption, and electrical performance analysis has been added to the “Control Unit Development” section.

And, the “Operational Methodology” contains the parameters such as actuation timing, picks per minute.

• Consider including CAD diagrams or exploded views of the module.

Author’s reply: Diagrams created using Adobe Illustrator have already been included in the submitted manuscript

5. Operational Methodology:

• Add visuals such as flowcharts or timing diagrams.

Author’s reply: The sequence of the steps in weft selection has already been discussed in the operational methodology.

• Move TINKERCAD simulation details to supplementary materials.

Author’s reply: It has been moved to supplementary materials (S1 Appendix).

6. Results and Evaluation:

• Benchmark your system against existing commercial devices (DORNIER, ITEMA, PICANOL).

Author’s reply: The primary objective of this research was to present a novel design, develop a functional prototype with improved weft selection capacity, and validate its operational feasibility. As the work represents a prototype-level development, it is not yet at a stage suitable for direct benchmarking against fully commercialized systems such as DORNIER, ITEMA, or PICANOL.

• Include more performance metrics: insertion time, miss-pick rate, energy use.

Author’s reply: Miss-pick rate has been discussed in the “Discussion and Implications” Section. And electrical performance analysis has been added to the “Control Unit Development” section. The device integration with the loom will be done as a future extension of this work. So, the weft insertion time, related data, is not available at present. The rapier is added as a prototype here to demonstrate the yarn gripping only.

7. Discussion:

• Excellent recognition of the rotor's role in angular stabilization.

Author’s reply: The authors appreciate the reviewer’s positive observation regarding the rotor’s contribution to angular stabilization.

• Add commentary on system scalability and industrial deployment potential.

Author’s reply: Additional discussion has been included on system scalability and the potential of industrial integration. The revised manuscript describes how this system can be integrated with the existing weaving machines and outlines how it can be scaled for broader applications.

8. Conclusion:

• Reiterate key performance outcomes.

Author’s reply: Operational speed, weft selection rate, pick insertion rate, miss pick rate, power requirements, and energy consumption have been discussed in the “Conclusion” section.

• Clarify next steps in industrial integration and synchronization with real loom cycles.

Author’s reply: The steps have been discussed in the above-mentioned section. After the loom integration, synchronization can be done as a further extension of this work.

9. References:

• Include more peer-reviewed references to increase academic credibility.

Author’s reply: Additional peer-reviewed references have been incorporated into the revised manuscript to enhance its academic credibility. Newly added references are reference no. 9, 11, 12, 19, 21, 24

Reviewer 2:

(1) How was the claimed 100% selection success verified over long production runs?

Author’s reply: A detailed report has been provided as a supplementary document titled “Weft Selection and Insertion Rate” (See S2 Appendix). It was prepared by conducting a test on 11 different yarns, each subjected to 5 trials using 20 feeders. A total of 1100 yarns were selected. This test provided more precise data on the weft selection rate (99.36%), pick insertion rate (99.27%), which are near 100%.

(2) Is there data on operational speed and any effect on loom throughput?

Author’s reply: The “Operational Methodology” contains the parameters such as operational speed, denoted as PPM (picks per minute), and actuation time. This device was designed as a prototype to show the design and its workability. The future development will be done on the loom integration, and the effect of the operational speed will be evaluated.

(3) Were yarn breakages or mis-picks quantified across yarn types?

Author’s reply: A supplementary document titled “Weft Selection and Insertion Rate” (See S2 Appendix), contains the miss-pick, which was 0.73%. 1100 yarns were taken to produce the test result.

(4) How stable is the angle control when the feeder spacing is further reduced?

Author’s reply: The algorithm can determine the feeders' angular position from the given weft pattern, and from this data, it synchronizes the module’s angular position, which is precisely controlled and held in position by the stepper motor. This motor can rotate 0.9 degrees/ step precisely.

(5) Why were no industrial weft selection benchmarks used for comparison?

Author’s reply: The primary objective of this research was to present a novel design, develop a functional prototype with improved weft selection capacity, and validate its operational feasibility. As the work represents a prototype-level development, it is not yet at a stage suitable for direct benchmarking against fully commercialized systems.

(6) Could pneumatic air pressure variation impact yarn quality over time?

Author’s reply: The maximum air pressure (1.27 bar ) in the circular tube is much less than that used in the Air jet looms (2.5 to 5 bar), and this air supply is provided to 20 feeders. Moreover, the device allows the air flow for a very short period of time during the weft catching, and only 3 inches of the yarn is subjected to air flow. So, it will not affect the yarn quality largely.

(7) How is the Arduino control synchronized with actual loom cycles?

Author’s reply: This will be done as future development, and its planning has been discussed in the “Discussion and Implication” section.

(8) Are there durability tests showing wearing of moving parts and guides?

Author’s reply: Though we tried to make the system with high-quality components but the durability largely depends on its electronic components, which have been discussed in the “Control Unit Development” section.

(9) What is the cost implication versus existing multi-selector systems?

Author’s reply: This device was developed as a prototype. It is not at the stage to be compared with commercial ones.

(10) Could vibration or electrical noise affect controller accuracy in real looms?

Author’s reply: This will be evaluated as part of the future development and Loom. Integration.

---

## [Decision Letter · Decision Letter 1]

25 Nov 2025

Development of a Mechatronic Weft Selector to Enhance Patterning Capacity in Rapier Looms

PONE-D-25-37019R1

Dear Dr. Mamun,

We’re pleased to inform you that your manuscript has been judged scientifically suitable for publication and will be formally accepted for publication once it meets all outstanding technical requirements.

Kind regards,

Ying Ma, Ph.D.

Academic Editor

PLOS ONE

Additional Editor Comments (optional):

Reviewers' comments:

Reviewer's Responses to Questions

**Comments to the Author**

1. If the authors have adequately addressed your comments raised in a previous round of review and you feel that this manuscript is now acceptable for publication, you may indicate that here to bypass the “Comments to the Author” section, enter your conflict of interest statement in the “Confidential to Editor” section, and submit your "Accept" recommendation.

Reviewer #2: (No Response)

2. Is the manuscript technically sound, and do the data support the conclusions?

Reviewer #2: (No Response)

3. Has the statistical analysis been performed appropriately and rigorously? 

Reviewer #2: (No Response)

4. Have the authors made all data underlying the findings in their manuscript fully available?

Reviewer #2: (No Response)

5. Is the manuscript presented in an intelligible fashion and written in standard English?

Reviewer #2: (No Response)

6. Review Comments to the Author

Reviewer #2: accept~~~~~~~~~~~~~~~accept~~~~~~~~~~~~~~~accept~~~~~~~~~~~~~~~accept~~~~~~~~~~~~~~~accept~~~~~~~~~~~~~~~accept~~~~~~~~~~~~~~~accept~~~~~~~~~~~~~~~accept~~~~~~~~~~~~~~~accept~~~~~~~~~~~~~~~accept~~~~~~~~~~~~~~~accept~~~~~~~~~~~~~~~accept~~~~~~~~~~~~~~~accept~~~~~~~~~~~~~~~accept~~~~~~~~~~~~~~~accept~~~~~~~~~~~~~~~

7. PLOS authors have the option to publish the peer review history of their article (what does this mean? ). If published, this will include your full peer review and any attached files.

**Do you want your identity to be public for this peer review?** For information about this choice, including consent withdrawal, please see our Privacy Policy .

Reviewer #2: No

---

## [Editor Report · Acceptance letter]

PONE-D-25-37019R1

PLOS ONE

Dear Dr. Al. Mamun,

I'm pleased to inform you that your manuscript has been deemed suitable for publication in PLOS ONE. Congratulations! Your manuscript is now being handed over to our production team.

Kind regards,

on behalf of

Dr. Ying Ma

Academic Editor

PLOS ONE